# Tableware trade in the Roman East: Exploring cultural and economic transmission with agent-based modelling and approximate Bayesian computation

Simon Carrignon[1,4,5]*, Tom Brughmans[2], Iza Romanowska[3¤]

**1** Center for the Dynamics of Social Complexity (DySoC), University of Tennessee, Knoxville, TN, United States of America, **2** Classical Archaeology and Centre for Urban Network Evolutions (UrbNet), Aarhus University, Aarhus, Denmark, **3** CASE, Barcelona Supercomputing Centre, Barcelona, Spain, **4** Department of Anthropology, University of Tennessee, Knoxville, TN, United States of America, **5** School of Information Science, University of Tennessee, Knoxville, TN, United States of America

¤ Current address: Aarhus Institute of Advanced Studies, Aarhus University, Aarhus, Denmark
* scarrign@utk.edu

**Data Availability Statement:** All relevant data are within the paper and its Supporting information files.

## Abstract

The availability of reliable commercial information is considered a key feature of inter-regional trade if the Roman economy was highly integrated. However, the extent to which archaeological and historical sources of inter-regional trade reflect the degree of economic integration is still not fully understood, a question which lies at the heart of current debates in Roman Studies. Ceramic tableware offers one of the only comparable and quantifiable sources of information for Roman inter-regional trade over centuries-long time periods. The distribution patterns and stylistic features of tablewares from the East Mediterranean dated between 200 BC and AD 300 suggest a competitive market where buying decisions might have been influenced by access to reliable commercial information. We contribute to this debate by representing three competing hypotheses in an agent-based model: success-biased social learning of tableware buying strategies (requiring access to reliable commercial information from all traders), unbiased social learning (requiring limited access), and independent learning (requiring no access). We use approximate Bayesian computation (ABC) to evaluate which hypothesis best describes archaeologically observed tableware distribution patterns. Our results revealed success-bias is not a viable theory and we demonstrate instead that local innovation (independent learning) is a plausible driving factor in inter-regional tableware trade. We also suggest that tableware distribution should instead be explored as a small component of long-distance trade cargoes dominated by foodstuffs, metals, and building materials.

**Funding:** TB received funding from The Leverhulme Trust ECF 2016-197 SC and IR received funding from the EPNet Project ERC-2013-ADG 340828 The funders had no role in study design, data collection and analysis, decision to publish, or preparation of the manuscript.

**Competing interests:** The authors have declared that no competing interests exist.

# 1 Introduction

## 1.1 Tableware trade in the Roman East

Vast quantities of foodstuffs, stones, minerals and craft products were traded over huge distances in Roman times, despite the significant limitations imposed by the then-current transport and communication technologies, and the uncertainties caused by climate, piracy and other factors. Seaborne commercial activity in particular facilitated long-distance trade flows throughout the entire Mediterranean region. However, the extent to which the commercial actors involved in this inter-regional trade could depend on abundant reliable commercial information about the supplies and demands of goods from other parts of the Roman world is still unclear [1]. This issue lies at the heart of current debates on the functioning of the Roman economy, where availability of reliable commercial information is considered a condition for the Roman economy to be highly integrated [2–5].

Archaeologists can gain insights into long-distance trade patterns and the availability of commercial information to ancient traders by excavating, documenting and studying ceramics. Their durability means they can be recovered in huge volumes, and the presence of non-local ceramics at sites can be an indication of long-distance commercial interactions [6, 7]. One of the most abundant and well studied sources of inter-regional trade flows is offered by imported ceramic tableware (non-local thin-walled fine ware plates, cups and bowls). They provide one of the most robust, well-studied and ubiquitous sources of data enabling comparison and quantification over centuries-long time periods, thus allowing for the study of the direction and intensity of trade throughout the Roman world [7, 8]. The recently aggregated tableware evidence from the eastern Mediterranean [9, 10] uniquely allows for the quantitative identification of centuries-long data patterns, revealing a particularly robust and well-studied distribution pattern [6] of different kinds of tableware between ca. 200 BC and AD 300. Here we present a computational model using this distribution of ceramic data to formally test hypotheses about the impact of the availability of commercial information to tableware traders (degree of market integration) on the distribution of the traded goods.

A large number of fine ceramic tablewares were produced in the eastern Mediterranean region during the late Hellenistic and Roman Early Imperial periods. Only a handful of these can be used to study long-distance trade because they achieved a commercial distribution that went beyond the immediate region around the center of production: so-called Eastern Sigillatas A-D (ESA, ESB, ESC, ESD). These four tablewares were produced in the eastern Mediterranean region, but a fifth western-produced ceramic tableware also achieved a very wide distribution in the Roman East: Italian Sigillata (ITS). The identification through surveys of these wares across hundreds of archaeological sites in the eastern Mediterranean allows us to observe trends in the width and overlaps of their distribution (their presence or absence at archaeological sites) for a period of five centuries between 200 BC and AD 300 (for detailed discussions of this distribution pattern, see [10, 11]).

This unparalleled tableware dataset has a high potential to contribute to a wide variety of studies of the Roman economy [8]. Here, we focus on how traders from one settlement access information regarding buying and selling strategies from traders in other settlements. Access to such commercial information would be beneficial in competitive markets, and the tableware evidence reveals in at least three ways how the market of imported tableware in the Roman east in this period might have been competitive:

1. The extremely wide distribution of ESA decreases sharply as soon as the western-produced ITS comes onto the eastern Mediterranean market (Fig 1, left panel) [10, 11].

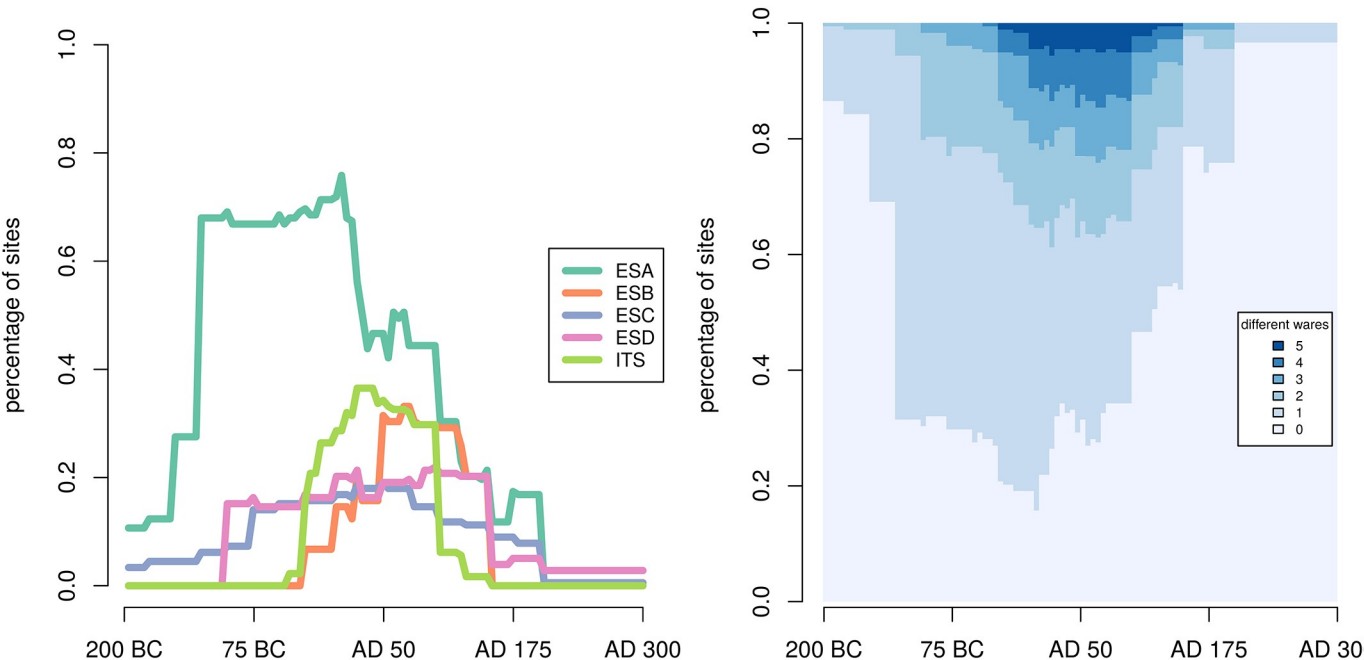

**Fig 1. Data patterns.** Data patterns derived from 8730 datable entries of 178 eastern Mediterranean sites from the ICRATES database. Left: percentage of sites with each pottery ware; note the disproportionate dominance of the earliest ware (ESA—teal), and its decrease with the introduction of the western-produced ITS (light green). We will later refer to this metric as *Pattern A*. Right: percentage of sites with a certain number of different wares. Note the dominance of sites with no or only one ware. We will refer to this metric as *Pattern B*. The two patterns capture two different aspects of the changes we are studying. Pattern A shows how each different tableware spreads to settlements throughout the time period under study, whilst pattern B shows how the diversity of tablewares changed from one settlement to another and over time. To ensure that the models of social learning we explore capture both aspects of these changes we test their ability to reproduce both patterns A and B at the same time.

2. Each tableware product was mostly distributed to settlements in the wider region around the production area, but there was significant overlap between the distribution areas of products [10, 12]. This suggests the eastern Mediterranean market was not divided into exclusive market zones of specific products: competing products serving a similar purpose could appear for sale in the same settlements (Fig 1, right panel).

3. The shapes of these ceramic vessels show clear morphological influences between products, especially the imitation of western-made ITS revealed by eastern-made vessels in the period 10 BC—AD 15. ESA vessel shapes were strongly influenced by ITS shapes [6, 13, 14], and the ITS practice of placing a stamp on the vessel with an inscription in Latin appears on ESB even though eastern-produced tablewares orignially never had these kinds of stamps [6, p. 54-59 15, 16] (for a detailed description of these three observations, see [17])

The processes that shaped these tableware distribution patterns and how they could reveal aspects of ancient inter-regional trade and the access to commercial information have been the subject of debates in archaeology and history (for a summary [10]). They could have been shaped by the production region being close to a large active urban hub with a productive hinterland, with plentiful availability of fuel, giving some wares a locational advantage over others [10, 18]. However, the ability to effectively deliver tableware to other large population centres or redistribution hubs might equally have been a factor [19]. Or the ability for a region's ceramic craft products to fill hulls on major shipping lanes of foodstuffs and building materials which made up the bulk of long-distance transport in the ancient Mediterranean [20], and which could have been state-driven. It has also been argued that the distribution pattern of

tableware in the eastern Mediterranean should be interpreted as evidence for limited availability of non-local commercial information due to the existence of different small-scale communities of traders trading particular tablewares, who disadvantaged outsiders trading other tablewares [5]. Previous formal modelling work has sought to explore some of these factors as potential explanations for the tableware distribution pattern: Hanson and Brughmans [21] explored the effect of the proximity of large population centres, whilst Brughmans and Poblome studied the effect of the availability to traders of local and non-local commercial information (where different wares were assumed to satisfy the same demand for tableware) [8, 22].

No doubt these tableware patterns need to be understood as the result of a complex mix of these factors, but it is outside the scope of our current paper to formally study all of these. Instead, in this study we focus on the question whether variable access (represented by social and independent learning processes) to non-local economic strategies can explain the tableware distributions. More specifically, we focus on the role of those traders who made decisions to buy and sell craft products over long-distances only, and who might have been influenced by each other's non-local buying and selling strategies. This perspective has never been formally studied as a driving factor for the explanation of the tableware distribution pattern, and we believe it to be complementary to previous formal modelling work. It further offers a new angle to explore the availability of commercial information in ancient inter-regional trade to contribute to ongoing debates on the degree of ancient market integration. Moreover, it is of particular research interest due to the stylistic influences between wares and the strong change in tableware distribution patterns when ITS was introduced in the eastern Mediterranean. We therefore explore the tableware market in the East as a competitive market where different wares were perceived as different products, and we evaluate whether this offers a credible explanation of tableware distribution.

## 1.2 Hypotheses

The archaeological information offers clear evidence of interaction among tableware traders in the eastern Mediterranean, suggesting a competitive market where economic strategies of those active in tableware trade could have influenced each other. It suggests the emergence of ITS in particular made the eastern tableware market increasingly competitive and triggered the producers and traders of eastern wares to change their practices.

What was the role of competition between traders from different settlements who bought and sold tableware in giving rise to these data patterns? Can the data be explained by traders from one settlement having access to buying strategies of traders from other settlements, despite the significant distances involved and the logistical limitations for people in the ancient world to gather reliable information? If so, does the copying of the strategies at the settlement where the traders are most successful in tableware trade offer a good explanation? Or were traders within one settlement not able to collect much reliable commercial information from those in other settlements, and did they instead change their commercial strategies independently or through chance encounters with other traders?

To explore those questions we translate them into three models of Social Learning Strategies that tableware traders might have employed:

1. independent learning: traders from one settlement independently change their tableware buying strategy (no access to reliable commercial information).

2. unbiased social learning: traders from one settlement randomly copy the tableware buying strategy of traders from another settlement (limited access to reliable commercial information).

3. success-biased social learning: traders from one settlement copy the strategy of traders from a more successful settlement (complete access to reliable commercial information).

By applying cultural transmission algorithms to the case study of Roman tableware production and trade we are able to test such assumptions about the importance of individual behaviours or the aggregation of behaviors of groups of individuals in shaping the political and economic history of the Roman World and other civilisations [23–26].

## 2 Materials and methods

### 2.1 Tableware data

The data used in this paper comes from the open access ICRATES database, the largest collection for the Roman East of excavated and published tableware fragments found at 275 sites across the eastern Mediterranean [9]. Here, we use a subset of 8730 chronologically datable ceramic evidences, considering only the presence or absence of the five wares (ESA, ESB, ESC, ESD, ITS) from 178 eastern Mediterranean sites: this represents the entire period of distribution in the Roman East for these wares. We only take into account the high level typological categories called wares in the archaeological literature (i.e. ESA, ESB, ESC, ESD, ITS) and do not distinguish between different versions of wares (e.g. ESB I-II, ITS-Arezzo) since this information is not coherently recorded between different excavations included in the database. Similarly, we only take into account the presence or the absence of a ware at a site. The quantitative volume of wares or the typological diversity recorded in the ICRATES database are not representative or comparable for sites with different sizes or proportions of excavated to not-excavated area or different strategies in recording and retention of ceramics. This is due to the typical practice in eastern Mediterranean Classical Archaeology to only publish diagnostic ceramic sherds: excavated ceramic counts or weights are rarely if ever published (this practice has been changing slowly in recent decades) [27]. The ICRATES database we use here is an aggregation of this practice for hundreds of sites, and it is not representative of the volume of past distributions. However, the presence or absence of the non-locally produced sigillatas we study here can be considered reliably recorded in this database. The practice of publishing diagnostic sherds dictates that at least the diversity of non-local wares is recorded and published, because these are very well-studied and can serve as chronological evidence, they would be well-known and readily identifiable by the excavators, and they serve as an indicator of non-local contacts and exchange. Moreover, the ware identification was checked and corrected where possible by the creators of the ICRATES database.

Each of these five tableware products was produced in a different region, typically by a number of production centres (Table 1). However, only for ESC production sites were excavated, in Pergamon and the surrounding region [28, 29]. Archaeologists were able to pinpoint the region of production of ESA in the Levantine coastal region between Latakia and Tarsos, of ESB in the Maeander valley in western Turkey and of ESD in (western) Cyprus, thanks to a combination of geochemical analyses and distributions of excavated pottery [11, 28, 30, 31]. ITS was produced in a range of western workshops among others in Arezzo, Pisa, Lyon and in the Po Valley.

The provenance of the data used is summarized in Table 1. The geographical distribution of the data is represented in Fig 2.

**Table 1. Typological, chronological references and possible region of production for major tablewares studied in this paper.**

| Ware | Abbreviation | Typological & chronological standard | Region of production [32] |
|------|--------------|--------------------------------------|---------------------------|
| Eastern Sigillata A | ESA | [31] | Coast between Tarsos (TUR) & Latakia (SYR) |
| Eastern Sigillata B | ESB | [31] | Maeander Valley in western Asia Minor (TUR); possibly Aydin (ancient Tralleis) |
| Eastern Sigillata C | ESC | [30, 31] & [28] | Pergamon & surrounding region |
| Eastern Sigillata D | ESD | [31] | Cyprus (probably the western part) |
| Italian Sigillata | ITS | [33] | Italy & Southern France |

To allow for identifying changes through time in distribution patterns shown in Fig 1, we draw on the standard dating ranges of the established typologies for these wares (Table 1). According to these, each morphological type has a different chronological date range. We count the number of sites at which each type of each tableware was found. We use cumulative probabilities to add up evidence at each site of the same ware but of different types with different dating ranges (some types have a narrow dating, but others can have very broad dating ranges). For each site/ware combination we calculate the probability that it existed in any given year, following the cumulative probability method—a well-established approach in Roman archaeology [10, 34, 35] and assuming a uniform probability distribution. To give an example, a pottery find that is dated between AD 1 and AD 10 will add the value of $\frac{1}{10}$ for each year between AD 1 and AD 10, because the probability that it existed in any one of those years

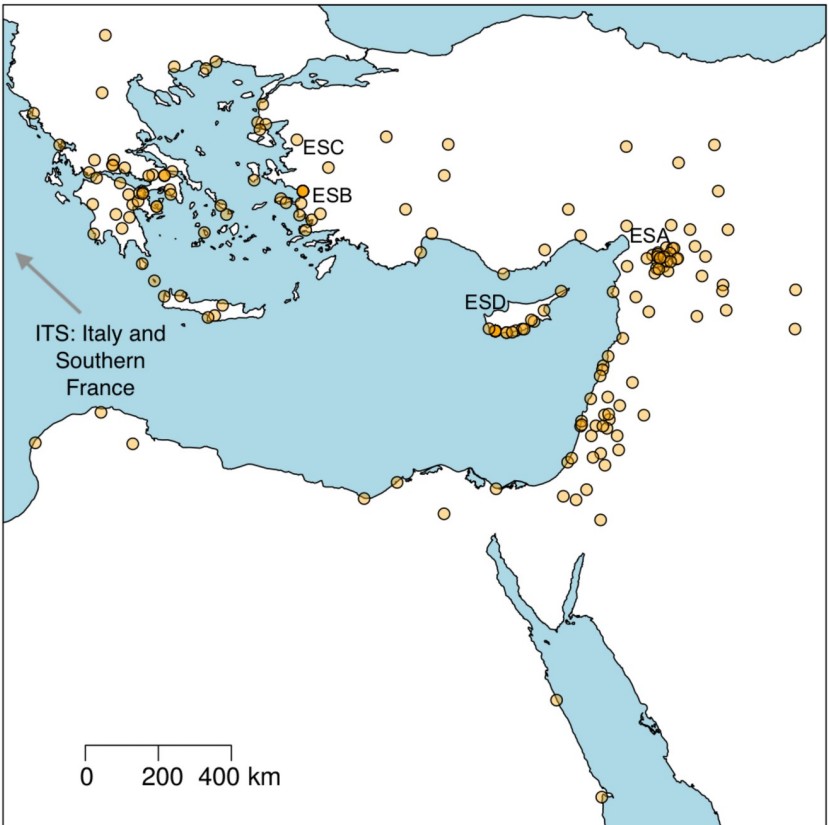

**Fig 2. Geographical distribution.** Geographical distribution of the 178 eastern Mediterranean sites from the ICRATES database. Each point represents one archaeological site. Presumed production regions of ESA, ESB, ESC, ESD are labeled. Map created using data from Natural Earth (http://www.naturalearthdata.com/).

is 10% assuming a uniform probability distribution. A pottery find dated to AD 1-100 will add the value of $\frac{1}{100}$ to each year between AD 1 and AD 100 because the probability that it existed in any one of those years is only 1%. As we add these partial probabilities together we get the cumulative probability that a given tableware was used at any one time point. As we keep on adding more pottery finds and we only explore whether wares were present or absent at a given point in time, then we reveal a chronological overview of the pottery distribution in the region (Fig 1).

## 2.2 Agent-based model

To implement our three hypotheses we use the agent-based model presented in full detail in [36]. This model builds on previous work by [37] and was re-purposed to combine cultural and economic aspects of trade, thus capturing how an agent can copy and learn from others within the context of trade activities. The copying of information and economic transactions are the two central elements of the model. In the model, agents copy cultural traits from other agents. These cultural traits are lists of values that agents attribute to the goods available in the environment. Agents will then use these lists to calculate prices to trade the different goods. In this paper, an agent is considered to represent the set of traders active in one urban settlement in the eastern Mediterranean, and theoretically able to trade with other sets of traders located at other settlements. The cultural traits these sets of traders exchange are here considered as the "economic strategies" of one settlement. These economic strategies spread through cultural interactions. This is the cultural component of the model and it is articulated around two mechanisms: social learning and innovation. The other central component, the economic component, is defined by three phases: production, consumption and trade. These mechanisms regulate how the goods themselves appear and travel in the environment. During the production phase, different agents produce different kinds of goods. In this paper the goods are different tablewares and money. During the trade phase, each agent (i.e. a set of traders at one settlement) brings their own good (one type of tableware or money) to a market where they try to exchange it to get all the other goods they did not produce. The way an agent exchanges their good against the others is defined by the cultural traits we mentioned above, represented by a list of values the agent attributes to each good. As we defined an agent as the set of traders at one settlement, this list can be seen as the aggregated commercial information of all the traders from one urban settlement. Then in the last economic phase, the consumption stage, agents consume (i.e. remove from their inventory) all the goods they collected during trade. This stage reflects the end of the inter-regional distribution of the goods and their deposition (through local trade and eventual disuse or breakage). After these stages, a score reflecting the success of the exchanges is given. In the setup designed for this paper, this score is built to reflect the ability to exchange enough but not too much ceramics of whatever ware. Algorithmic descriptions of those steps are given in the Algorithms 1 and 2.

An even more detailed description of the model implementation and a thorough theoretical exploration are available in [36]. More details on the economic properties of the model can be found in the original paper by [37] and the relationships between economics properties and different learning strategies have also been studied in [38]. Here we extend those studies by testing the model against empirical evidence.

In short, we use an adjusted model from [36] to answer our substantive questions about trade in the Roman East 1.2, which are at the core of debates in Roman economy studies, using a large dataset of ceramics 2.1. The model was therefore adapted to reflect the historical and archaeological context described in section 1.

The model simulates the period of 500 years, from 200 BC to AD 300. The order of introduction of different tablewares to the market follows the widely accepted chronology (shown in Table 1; Fig 1):

**From 200 BC to 101 BC**: ESA, ESC

**From 100 BC to 41 BC**: ESA, ESC, ESD

**From 40 BC to 28 BC**: ESA, ESC, ESD, ITS

**From 27 BC to AD 149**: ESA, ESB, ESC, ESD, ITS

**From AD 150 to AD 199**: ESA, ESC, ESD

**From AD 200 to AD 300**: ESC, ESD

New products are introduced to the market at simulation times equivalent to this chronology (cf. Section 3.2).

We simulate 500 agents, where each agent represents the aggregated economic activity of the traders from one urban settlement. This number roughly approximates the number of urban settlements in the Roman East estimated by [39, 40] (cf. Section 3.2).

The original model [36] had an evenly distributed ratio of producers vs consumers, which in this implementation was modified to five producers (one for each type: ESA, ESB, ESC, ESD, ITS) and 495 consumers.

In addition to these five products, money was introduced as a sixth good used by agents to exchange for other products (the special status of money and its impact on the original model is discussed in [37]).

The algorithmic description of the model is given in Algorithm 1, where the functions that introduce or remove goods from the tableware market, *historicalChange*() and *update*(), follow the chronological sequence described above.

The three hypotheses described in Section 1.2 are implemented in the following way:

1. Independent learning hypothesis: the call to *CulturalTransmission* in line 17 is removed. No cultural transmission takes place.

2. Unbiased learning: a selection mechanism randomly selects cultural traits from the population with a probability $\mu$ instead of *CulturalTransmission*.

3. Success-bias mechanism: the *CulturalTranmission* process is performed as described in Algorithm 2.

**Algorithm 1** Model's algorithmic structure.

```
1: INITIALIZATION:
2: for i ∈ #Pop do          ▷ Initialize the agent with no goods and a
random value vector
3:    Q^i = (0, ···, 0)
4:    V^i = (v^i_0, ···, v^i_n)        ▷ The values of v^i_j are selected randomly
5: SIMULATION:
6: loop step ∈ TimeSteps
7:   if historicalChange() then
8:     update(Q, V)          ▷ We add or remove a product given archaeo-
logical evidence and update associated values
9:   for i ∈ Pop do
10:     Production(Q^i)
11:   for i ∈ Pop do
12:     for j ∈ Pop do
13:       TradeProcess(V^i, Q^i, V^j, Q^j)
```

```
14:    for i ∈ Pop do
15:      ConsumeGoods(Q^i)              ▷ All goods are consumed
16:      if (step mod ω) = 0 then
17:        CulturalTransmission(V)
18:        Innovation(V^i)
```

**Algorithm 2** Cultural Transmission Process.

```
1: ToGet = 0.2 × (#Pop)/(#Good)
2: for g ∈ Good do
3:    ToReplace = {}
4:    while #ToReplace < ToGet do
5:      j = SelectRand(Pop, g)         ▷ Select randomly an agent j
among the agents producing g
6:      X ∼ U([0, 1])          ▷ Draw a random number from the uniform
distribution between 0 and 1
7:      if X > ComputeScore(j) then        ▷ Select preferably the
agents with the lowest scores
8:        ToReplace = {ToReplace, j}
9:    while #ToReplace > 0 do
10:     j = SelectRand(ToReplace)
11:     i = SelectRand(Pop, g)         ▷ Select randomly an agent i
among the agents producing g
12:     X ∼ U([0, 1])
13:     if (X < ComputeScore(i)) then        ▷ Select preferably an
agent i with a high score
14:       if (ComputeScore(i) > ComputeScore(j)) then        ▷ Verify
that agent i has a higher score than agent j
15:         CopyPrice(i, j)
16:         ToReplace = ToReplace − i
```

We summarized the model's parameters and their initial values where relevant in Table 2. Some values are fixed and others, noted with $\mathcal{S}$, are randomly sampled and explored via Approximate Bayesian Computation (ABC). Note that some parameters are not used in all models (for example λ, the rate of social learning, is not used in the independent learning model).

## 3 Approximate Bayesian computation

To compare and test the robustness of the results of different models and scenarios against empirical data we use the Bayesian inference paradigm. Bayesian inference allows to associate

**Table 2. Model parameters.**

| parameter | description | initial value |
|---|---|---|
| $t$ | Total number of economic interactions | $\mathcal{S}$ |
| $\omega$ | number of economic interactions per cultural interaction | $\mathcal{S}$ |
| $CI$ | total number of cultural interactions | $\mathcal{S}^*$ |
| $\mu$ | rate of innovation | $\mathcal{S}$ |
| $\lambda$ | rate of social learning | $\mathcal{S}$ |
| N | total number of agents | 500 |
| $\mu_{max}$ | variance of innovation | $\mathcal{S}$ |
| $\lambda_{str}$ | strength of bias (when social learning is biased) | $\mathcal{S}$ |
| $n_{good}$ | number of types of goods (e.g. ESA, ESB, ...) produced and exchanged | 3-6 |

Note that some parameters are not used in all modelled scenarios. Parameters with initial value $\mathcal{S}$ will be explored via Approximate Bayesian Computation and thus randomly sampled from distributions (cf. Section 3.3 and Table 3). *CI is not an explicit parameter of the model but is defined in relation to other parameters as $CI = t \times \omega^{-1}$.

a probability value that a hypothesis is true given the evidence. The hypothesis can be represented as any quantifiable and measurable model, while the evidence can be any empirically measured distribution. In short, Bayes theorem

$$P(A|B) = P(B|A) \times P(A)/P(B)$$

allows to compute a probability of a hypothesis being true given the known evidence.

$P(A|B)$—the posterior: the probability that a model $A$ is true given evidence $B$.

$P(B|A)$ are all possible outputs of the simulation.

$P(A)$—the prior: the initially assumed probability distribution.

$P(B)$ is the distribution of the evidence.

It is possible to iteratively update those probabilities when new evidence becomes available, and use that new evidence to update the previously calculated posterior. Thus, Bayesian inference can be used to weigh different hypotheses against the same empirical data set, allowing for the identification of the hypothesis that has the highest probability of being true given the data. Its potential for archaeological research has long been recognised [41], but the number of applications remains limited.

An important obstacle to the generalisation of this method is that the narrowing of the posterior probability to meaningful ranges is often impossible. This is especially the case when the mechanisms behind the pattern are unknown, when they are highly stochastic, and when their description depends on multiple theories from different fields, as in the current study. However, dramatic increases in the available computational power and the use of randomized simulations have led to the development of an innovative way to overcome this problem: Approximate Bayesian Computation [42] (ABC). This technique relies on the random generation of the prior values to simulate the full parameter space of the hypothesis and to evaluate the resulting posteriors in light of the available data. The great potential of the combination of cultural evolution modelling and ABC has been widely recognized [43–46]. Nonetheless, the probabilistic and stochastic nature of this method implies that one needs to perform a very high number of simulation experiments, in the magnitude of hundreds of thousands, which is a challenge when the model explored is as complex and computationally costly as the one presented here.

## 3.1 Population Monte Carlo

[47] proposed a solution to optimize the number of simulations needed to approximate the posterior distribution in ABC, known as Population Monte Carlo (ABCPMC). This method accelerates the computation of the posteriors by updating the priors used throughout the process and dynamically lowering an $\epsilon$-threshold used to consider a simulation as close enough to the data.

We briefly summarized the mechanics of ABCPMC in algorithm 3, and here we provide a more elaborate description of the approach. First of all, and just like any other ABC technique, it needs a prior distribution that describes the space where the parameters of the model can be chosen. This prior, that we later describe in Table 3, is composed of various distributions from which we initially randomly sample the parameters of our model. As these distributions are mostly uniform, this often simply consists of randomly choosing a number between a given interval, with the interval being informed by our knowledge of the system. This is repeated for all the parameters of the model, and it results in a combination of parameters that we can use to run the model. Several combinations are generated, and the corresponding simulations are run.

**Table 3. Prior distributions for parameters to be inferred by the ABC.**

| Parameters | Priors | Description |
|---|---|---|
| $\mu$ | $U(0, 1)$ | rate of innovation |
| $\mu_{max}$ | $U(0, 10)$ | variance of innovation |
| $\lambda$ | $U(0, 1)$ | rate of social learning |
| $\lambda_{str}$ | $U(0, 10)$ | strength of social learning bias |
| $t$ | $U^*(50, 1000)$ | total number of economic interactions |
| $\omega$ | $U^*(1, 50)$ | number of economic interactions per cultural interaction |

$U(X, Y)$ corresponds to the uniform distribution between $X$ and $Y$.

$^*$To generate plausible simulations the total number of cultural interactions needs to respect the constraints described in the text, thus some combinations of $t$ and $\omega$ have to be rejected.

The result of those simulations are then compared to the data. In this study, this comparison is done by the function later described in Eq 2. In a classical ABC algorithm, the result of this function is compared against a (unique) value $\epsilon$, defined as the threshold under which a simulation is considered as reproducing the data. All combinations of parameters that did not generate results falling under the threshold $\epsilon$ are rejected. The remaining combinations of parameters are said to draw an approximation of the posterior distribution of the model, i.e. the combinations of parameters that allow to reproduce the observed data.

As stated above, the challenge with such a rejection algorithm, as the one used in [43], is that they often need a huge number of simulations to generate enough combinations of parameters under the threshold $\epsilon$. This is why an ABCPMC variation is used. In the ABCPMC presented here, a series of decreasing $\epsilon_i | i \in 1..s$ is defined. For the first level, $\epsilon_1$ is set to be high enough to reject only parameters generating results very far from the data. Then, instead of using the combinations of parameters accepted at this stage as the posterior distribution of our model, we use the information given by this "first step posterior" to inform the prior of the next step.

What happens in practice is that instead of sampling uniformly the full parameter space, we use the results of the first selection to get some intuition of the areas of the parameter space that are very unlikely to produce good simulations. Then we use these probabilities to generate new combinations of parameters that will be tested in a second step. In other words we use the posterior distribution of one step to inform the prior distribution of the other step. This way, we avoid testing the model again and again with parameters that give obviously wrong results. In parallel to this dynamical updating of the priors, the value of the threshold $\epsilon_2$ used in the second step is chosen to be lower than $\epsilon_1$.

The process is then repeated, the combination of parameters that generated results below $\epsilon_2$ are selected and this selection is used to inform the sampling of a third generation of parameter combinations, parameters that will be used during a third step with again a lower threshold $\epsilon_3$. And so on, until the posterior do not change anymore from one step to another, or until the time to generate enough simulations below a threshold $\epsilon_e$ becomes too long. The combinations selected during the last step are considered to be the posterior of our model. This method not only allows to avoid testing a huge number of useless simulations and thus saves a lot of time, but it also allows to explore ranges of parameters outside the prior interval definition by dynamically updating the space explored.

We built our own version of ABCPMC (Algorithm 3) based on the ABCPMC algorithm described by [47] and developed upon the Python implementation proposed by [48]. It is optimised to work with our model and a supercomputing environment.

## 3.2 Summary statistic and distance to data

One of the key elements to all ABC approaches is the function used to calculate the distance to the data (the $\Delta(s, x)$ at line 10 in Algorithm 3) and how the data ($x$) and simulations ($s$) are represented. The nature of the data used here raised some problems leading to the development of a bespoke method for calculating the summary statistics.

In the following paragraphs we describe the sampling and normalization procedures performed on the empirical data to derive summary statistics. We pre-processed both patterns (A and B) using these procedures.

**3.2.1 Summary statistics.** *Site sampling*: The archaeological dataset includes 178 sites (cf. Fig 2). However, the estimated number of urban settlements in the Roman East [39, 40, 49] is closer to 500. The model uses the latter figure. In order to compare the population of 500 agents with the available 178 observations we normalize both metrics (patterns A and B in Fig 2) as a percentage of the total. As a result we compare the percentage of sites and wares rather than their absolute number. This assumes that the proportions described in the dataset are representative of the proportions we would find if we had access to all urban settlements active during the period studied.

**Algorithm 3** ABC: The Population Monte Carlo algorithm

```
1: INITIALIZATION:
2: ε = GenerateEpsilons()          ▷ Generate a set of decreasing εs
3: θ₁ = GeneratePrior()
4: RUN:
5: for εₜ in ε do
6:    while pool.size < 500 do
7:       θᵢ = prior.genNewParam(εₜ)   ▷ Draw a vector of parameters from
the prior
8:       r = Model(θᵢ)          ▷ Simulate the model
9:       s = summary(r)         ▷ Generate summary statistic
10:      if Δ(s, x) < εᵢ then
11:         pool.add(θᵢ) ▷ Add to the list of parameters used to update
the prior
12:      else
13:         rm(θᵢ)
14:   prior = ModifyPriors(pool) ▷ Modify the prior using selected θs
covariance matrix
   return pool
```

*Time binning*: To calculate the number of different types of ware present in one site at one time period, a duration (in years) has to be defined for this time period. The dataset can then be divided into a finite number of periods that can be used to divide the results of the simulation in a similar way. If we choose to split the data into two periods, then the model will simulate two different periods and compare them to the data, if we decide to split the dataset into 100 periods, then the model will simulate 100 periods and the ABC will compare each one of them with the real data. On the one hand, having more periods makes the comparison between the model and the data more time consuming. On the other hand, by splitting the data in too small a number of time periods, the patterns described in Section 2.1 disappear. The impact of binning the data with periods of different sizes is represented in Fig 3. With 10 bins the described properties of the data pattern disappear to be replaced by straight lines. In contrast, binning the data into between 50 to 200 periods does not change the overall trends. To strike the right balance between loss of information (less bins) and computation cost (more bins) we divided the dataset into 50 periods of 10 years as it preserves the data pattern sufficiently. In the model, a period is defined by the number of cultural interactions, i.e., opportunities for agents to copy from each other: every simulation is split into 50 periods composed of an equal

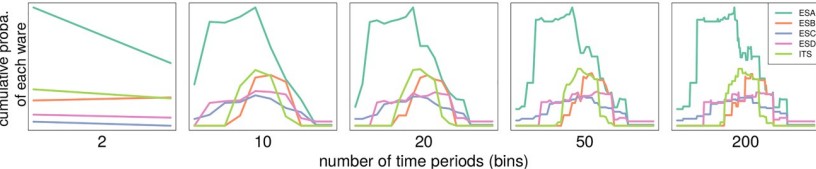

**Fig 3. Impact of the number of bins on the data pattern.** The number of bins used to describe the data increases from left (2 bins) to right (200 bins).

number of cultural interactions. The total number of cultural interactions per period is inferred by the ABC.

**3.2.2 Distance to the data.** Although multiple distance functions were tested to compare the data to the model output, we decided to use here an adapted version of the Euclidean distance between the data and the simulation at each period and for each measurement, as in [43] (see Eq 1).

$$\delta(s, d) = \sqrt{\sum_{p=1}^{P} \sum_{i \in W} (s_{i,p} - d_{i,p})^2 \times \frac{1}{P \times W}} \tag{1}$$

Where $d$ is a proportion measured with the real data, $s$ a proportion measured with the simulated data. $W$ is the set of categories for the pattern observed: the five different wares in the case of pattern A ($W$ = {ESA, ESB, ESC, ESD, ITS}), and the six possible levels of diversity in the case of pattern B ($W$ = {0, 1, 2, 3, 4, 5}). $P$ is the number of time periods that the dataset and the simulation have been split into ($P$ = 50). We measure proportions as described in the previous section, thus for pattern A: $d_{i = ESA, p = 2}$ represents the percentage of sites with ESA during the second time period, while for pattern B: $d_{i = 3, p = 2}$ represents the percentage of sites with 3 different tablewares during the second period. The same applies for the proportions found in our simulation $s$. Given this, if 10% of the sites have none of the 5 tablewares during the second period of a simulation ($s_{i = 0, p = 2}$ = .1), as we measure in the real data that this percentage should be 86.5% ($d_{i = 0, p = 2}$ = .865), the distance will be $(d_{i = 0, p = 2} - s_{i = 0, p = 2})^2 = (0.865 - .1)^2 = 0.585$. We apply this calculation to all wares and for all periods, and we combine the two patterns A and B by taking the mean between the two scores. This gives us a global metric measuring how far our simulations are from the real data; a metric that takes into account both patterns A and B as summarized by Eq 2.

$$\Delta(s, d) = \frac{\delta(s_A, d_A) + \delta(s_B, d_B)}{2} \tag{2}$$

### 3.3 Parameters & prior distributions

Prior distributions were selected to cover wide but historically credible ranges of parameters (Table 3). The rates of innovation ($\mu$) and social learning ($\lambda$) are sampled from a uniform distribution between zero and one. The number of economic interactions $t$ (i.e., buying and selling) is chosen to be between one and three interactions per year. The prior for the number of economic interactions per cultural interaction ($\omega$) is also randomly sampled but needs to respect different constraints: at least two economic interactions take place between every two cultural interactions to allow for information to be gathered ($\omega >= 2$), there are a maximum of two cultural interactions per year ($\frac{500}{CI} <= 2$) but at least one cultural interaction per period ($CI >= 50$).

**Table 4. Value of epsilon for all steps of the ABCPMC.**

| step | 1 | 2 | 3 | 4 | 5 | 6 | 7 |
|---|---|---|---|---|---|---|---|
| $\epsilon$ | 0.13 | 0.011 | 0.0109 | 0.0108 | 0.0107 | 0.0106 | 0.0105 |
| step | 8 | 9 | 10 | 11 | 12 | 13 | |
| $\epsilon$ | 0.0104 | 0.0103 | 0.0102 | 0.0101 | 0.0100 | 0.0099 | |

In addition to the parameters of the model, the ABC algorithm itself takes as input a decreasing sequence of $\epsilon$s (Table 4).

## 4 Results

We ran the ABC Algorithm 3 for 13 different $\epsilon$s decreasing logarithmically (cf. Table 4) for each of the tested hypotheses: Success-biased Social Learning, Unbiased Social Learning and Independent Learning (cf. Algorithm 1). Each step of the ABCPMC was completed when 500 simulations produced results deemed acceptable. A simulation is considered acceptable when its distance to the data $\Delta(s, d)$ (where $s$ are the simulated results and $d$ the data) falls under the acceptance level $\epsilon$. To reach the required number of simulation runs (6, 500 in total: 13 steps x 500 runs) 206, 902 simulation runs were required for the Independent Learning model, 564, 211 for the Unbiased Social Learning model and 1, 267, 560 for the Success-biased Social Learning model. The percentage of simulation runs meeting the $\epsilon$ is shown in Fig 4. It shows that the ratio of simulations matching the data to a satisfactory degree is significantly lower for Success-biased Social Learning than for the other two hypotheses. Among them the

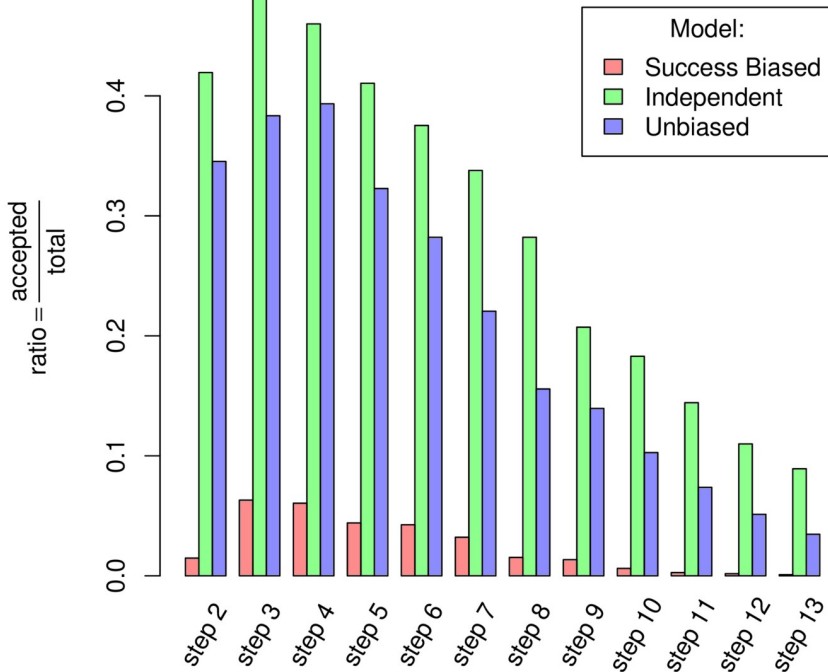

**Fig 4. ABC accepted simulation ratio.** Evolution of the ratio between the number of accepted simulation runs (ie simulations generating output $s$ where $\delta(s, d) < \epsilon_{step}$) and the total number of simulations needed to accept 500 simulations at each step of the ABC algorithm and for the three models. The first step has been removed as it represents an $\epsilon$ big enough to accept any simulation run, which leads to a ratio of 1 for all models.

Independent Learning model scores consistently higher than the alternative Unbiased Social Learning model.

## 4.1 Model selection

Using the approximation of the likelihood calculated by taking the simulations selected at the last step of the ABC procedure, we compute the Bayes Factor for all models to formally quantify their relative probabilities [50, 51].

We will note $K$ the Bayes Factor between model $m_1$ and $m_2$ as:

$$K_{m_1,m_2} = \frac{P(D|m_1)}{P(D|m_2)} \tag{3}$$

Where $P(D|m_i)$ is the likelihood of model $m_i$, as estimated by the ABC given data $D$.

Independent Learning is the mechanism explaining the data best (Table 5). It is 1.96 times more likely than Unbiased Learning and is approximately 23.5 times more likely than the Success-biased Social Learning hypothesis [52]. Thus, the Independent Learning model produces simulations matching the data more frequently and their output is closer to the empirical data pattern than either of the other models. These Bayes factors by no means give an absolute metric that ultimately validates the Independent Learning model. They illustrate that, given the evidence we have and the models we compared, the Independent Learning model is the more likely to reproduce the data observed.

## 4.2 Posterior distributions

The ABC allows us to calculate the posterior distribution of a model described by the parameters $\theta = [\theta_1, \ldots, \theta_p]$ given the data $d$: $P(\theta|d)$. This posterior distribution gives us the parameter combinations that allow our model to reproduce the data, and at the same time which among those different combinations are most likely to do so. By comparing the marginal posterior distribution of each parameter $\theta_1, \ldots, \theta_p$ to the priors we described in Table 3 we can see which are the important parameters as they will have the narrowest posterior as compared to the prior. The most significant of these posteriors have been graphically represented in Fig 5 (see S1 Fig for all).

The posterior distributions of all parameters for the Independent Learning and the Unbiased Social Learning models are very similar (green and blue, Fig 5). The only exception is the rate of social learning, which is not used in the Independent Learning hypothesis thus staying equal to the prior distribution. The fact that the rest of the posteriors for both models are similar is in line with the Bayes factors shown in Table 5 where those two models have a ratio close to one.

**Table 5. Summary of the Bayes factors between all models.**

|  | Unbiased | Independent | Success Biased |
|---|---|---|---|
| Unbiased |  | 0.5 | 12 |
| Independent | 1.96 |  | 23.5 |
| Success Biased | 0.08 | 0.04 |  |

This table should be read row by row: the number in each cell reveals how much more likely the model in the corresponding row is to explain the data than the model in the corresponding column.

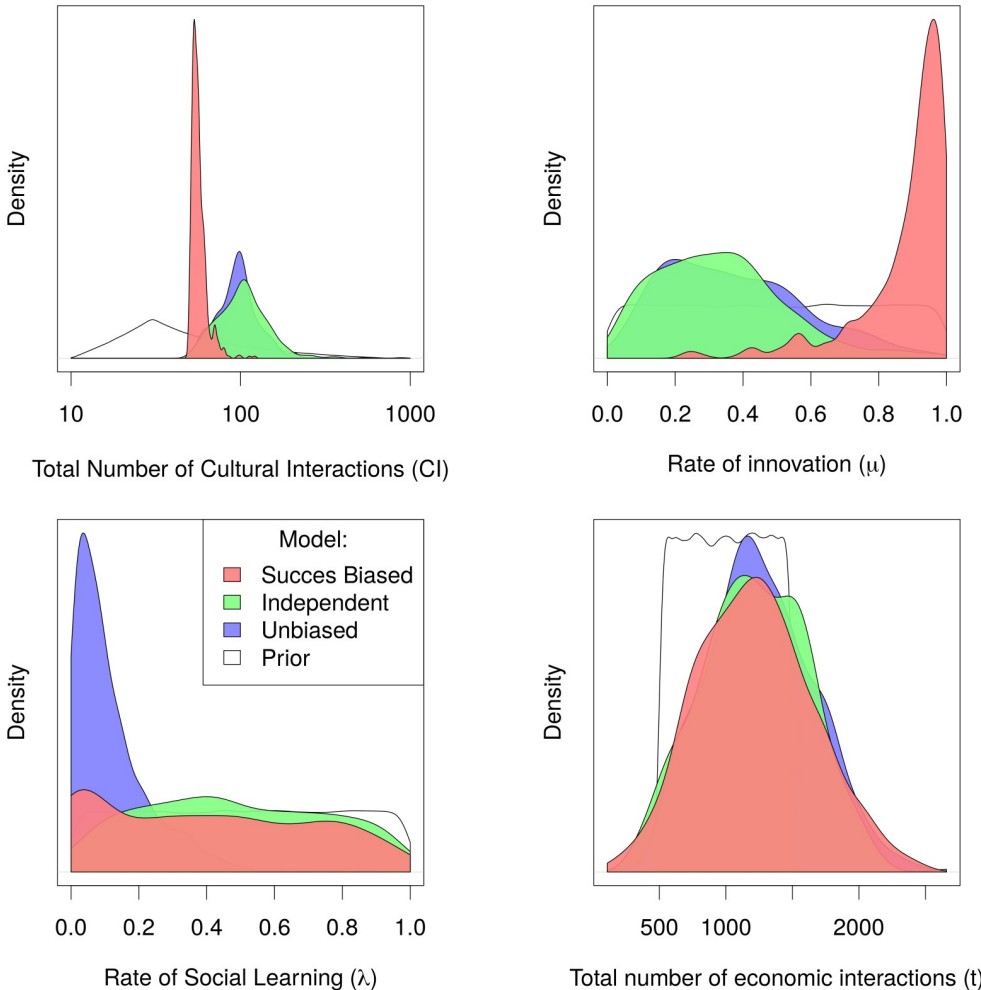

**Fig 5. Posterior distributions for the three models.** Posterior distributions of parameters drawn using the 500 accepted simulations from the last ABC step ($\epsilon = 0.0099$).

The posterior distributions indicate what range of parameter values matched the data best across all runs of the Independent Learning model. The high density regions (HDR) represent the parameter space within which 75% and 95% of acceptable simulations lie (Fig 6).

The HDRs of posterior distributions of key parameters offer a glimpse into the circumstances under which the observed data pattern might have formed.

**Total number of economic interactions**. The number of times agents go to the market to buy tableware during the whole simulation. The 75% HDR falls between 750 and 1700 economic interactions, i.e. 1.5 to 3.4 times per year.

**Total number of cultural interactions**. The number of times agents had the opportunity to copy strategies from other agents. The 75% HDR falls between 63 and 140 cultural interactions, i.e. once every 7.9 to 3.6 years.

**Rate of innovation**. The probability that an agent changes its strategy independent of other agents. The 75% HDR falls between 8.8% and 51%.

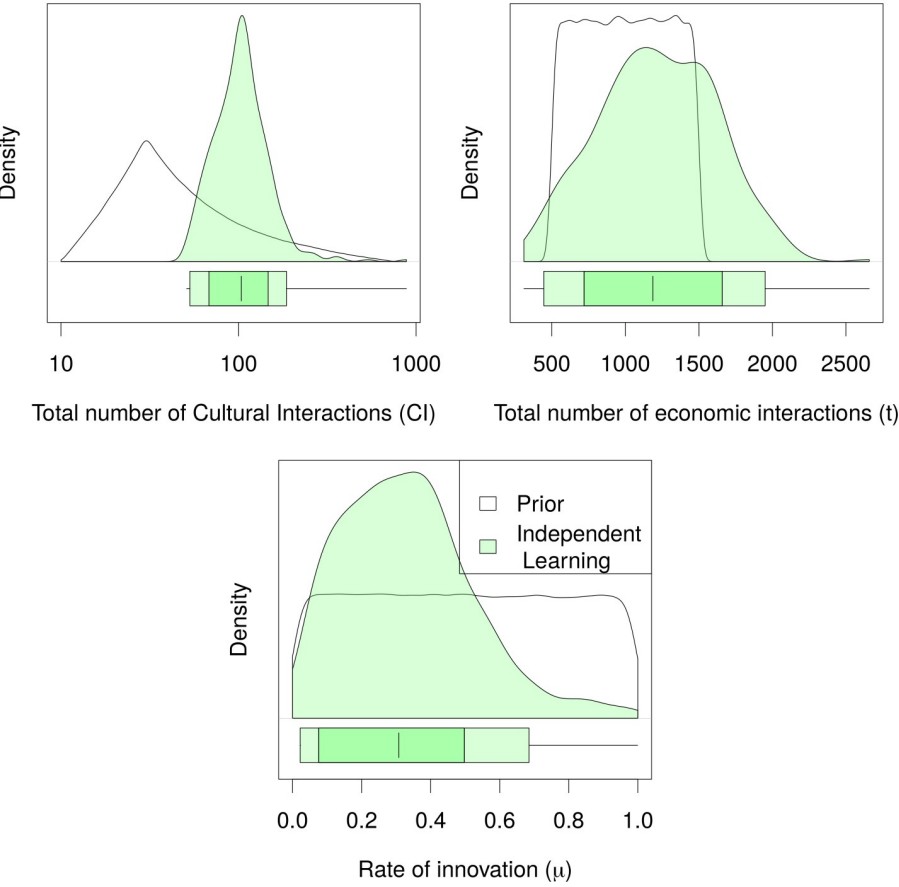

**Fig 6. Posterior distribution and high density regions for the model of independent learning.** Marginal posterior distributions of the independent learning Model's parameters. The boxplots at the bottom of each graph shows the 75% HDR (darker green) and the 95% HDR (lighter green). The vertical line indicates the mode of the distribution.

## 5 Discussion and conclusions

### 5.1 Tableware trade in the Roman East

Ceramic tableware data patterns reveal clear influences between different wares (distribution patterns; stamps; vessel shapes) as revealed by the archaeological data (section 2.1), reflecting a competitive market. Nonetheless the results of our simulation, in which cultural copying mechanisms were intertwined with economic mechanisms suggested that the economic strategies of sets of traders from one settlement were independent from the strategies of sets of traders from other settlements.

Based on the comparison of results of the three hypotheses with the archaeological data using ABC, we observe that the model, in which commercial agents change their tableware buying strategies independently generates simulations matching the archaeological data patterns better. The two other models, where agents update their strategies by copying strategies from other agents, were less able to match the data.

This result has important implications for the study of ancient inter-regional trade in the eastern Mediterranean, and for Roman economy studies more broadly. It shows that copying the strategies of successful sets of traders at settlements within the tableware market may not have played a central role in inter-regional tableware trade: copying successful strategies which

implies a high level of information flow clearly does not describe well the nature of this market. Independent and locally-oriented tableware economic strategies seem to be more plausible processes of tableware trade in the Roman East.

Another reason could be that long-distance distribution of tableware was intertwined with major trade flows from regions with foodstuff production (e.g. present-day Tunisia and Egypt), mining and quarrying activities (e.g. present-day southern Spain and the Egyptian eastern desert): tableware was one of the additional cargoes to fill up empty spaces in ships' hulls [20]. Thus, the mechanisms for driving inter-regional trade in other goods such as foodstuffs, stone and metals may offer better explanations than those considered here for just tableware trade. These foodstuff and mineral products made up the bulk of all long-distance trade in the Roman world, and their study is key to understanding the Roman economy. If the long-distance distribution of tableware was entirely structured by the long-distance trade flows in these other goods, then we would indeed expect the tableware evidence to suggest that independent learning is likely on the part of traders active in long-distance tableware trade. These traders would have had no advantage in obtaining each other's economic strategies and reacting to them, because they only had access to the ceramics that happened to travel along with other goods. However, this still leaves open the option that access to commercial information about tableware and about other goods were linked: that long-distance trade might have relied on access to both and not only on access to information about tableware as tested in this study.

We believe our results highlight the need for future work to focus on the link between long-distance trade in different goods. Doing so would require several large open access comprehensive datasets of centuries-long amphora container or stone distribution data for the entire Roman Empire or significant parts of it. However, we believe it to be crucial for such studies to make use of the formal modelling methods used here, given that the dependency on each other of different goods' long-distance trade and distribution can be conceptualised in many ways. By quantifying a range of mechanisms for trade in craft products, our study presents a pipeline for studying this key topic in Roman archaeology and history using archaeological data. The precise nature of the link between trade in craft products and other goods and the use of cultural/economic transmission models with other types of evidence of inter-regional trade such as distributions of amphora containers and raw materials such as stone should be explored in future work.

We should equally consider that the assumption used here that each ware was a commodity for which a distinct demand existed is not appropriate. Perhaps the sharing of information about different wares did not matter, because each ware satisfied the same demand? Indeed, previous formal modelling work that did not share this assumption (but focused on access to both local and non-local commercial information) came to the conclusion that the tableware distribution was structured by a relatively high degree of access to commercial information [8].

Interpreting the posterior distributions of model parameters allows us to explore the implications of the independent learning hypothesis in much more detail: they offer realistic parameter ranges for specifying theories concerning the economic and cultural interactions. Although these credible parameter ranges are wide, this is not unexpected in a study of an economic system that functioned two millennia ago where very little information is available that would enable narrowing prior distributions of parameters. However, they still allow us to add an, for Roman economy studies, unprecedented level of specification to the theory of independent economic innovation. In acceptable simulations the traders of one settlement bought tableware from those at other settlements around 1.5 to 3.4 times per year (75% HDR). In our model this refers only to the long-distance interactions between traders, and not the frequency of subsequent local sales of the non-local ceramics. Such limited frequencies for long-distance

interactions are certainly historically and archaeologically supportable, given the significant limitations on the frequency of obtaining products from other parts of the Empire posed by the then-current transport technologies and the financial requirements to organise inter-regional shipping. Oversea travel between ports in the eastern Mediterranean took days and travel from the Eastern to the Western Mediterranean implied by the distribution of ITS could take weeks. For example, Warnking's simulation study of the sailing time from Puteoli in Italy to Alexandria in Egypt suggests this would take at least one week under ideal conditions, and he argues this is similar to the travel time stated by Pliny the Elder for this trip [53]. But sailing conditions and the associated risks varied heavily depending on the season. In this context, assuming long-distance economic interactions between traders based at hunderds of sites throughout the Mediterranean was possible 1.5 to 3.4 times a year seems plausible. However, the suggested rate at which traders within one settlement updated their tableware buying strategies is far more limited, ranging from every 1.6 years to every 16 years (75% HDR). So even though independent changes of strategies are the most plausible scenario, these changes simply did not seem to have occurred very frequently. To remain closer to the historical context, future studies should focus on comparing these time estimates with what we know about ancient transport and communication infrastructure and technology [54–56], exploring the implications of this theory for the mobility and activity of commercial actors active in inter-regional trade in craft products in late Hellenistic and Roman times.

Our results revealed little about the role of ITS and we believe this western-produced ware should be the focus of future computational modelling research. Although we performed experiments to specifically explore how the presence of ITS might have stimulated competition on the eastern market (see SI S2 Fig), our approach did not succeed in identifying any effects other than those presented above. This should be further explored in future studies, alongside exploring whether success-bias might be a particularly viable theory for the much shorter time period 40 BC to AD 150, when ITS spread across the eastern Mediterranean. We also believe that the nature and processes of possible stylistic imitation of ITS features by eastern wares should be explored from a cultural transmission perspective using the methods we applied in this study.

## 5.2 Methodological insights into linked cultural economic processes

Our results suggest direct success-biased social learning does not offer a good explanation for the phenomena we studied. However, different forms of social learning (unbiased, prestige-biased, success-biased, content-biased, etc.) might well have more explanatory value in other research contexts. To facilitate these potential applications we consider a number of methodological points revealed by this work.

The first obstacle to the empirical detection of a direct success-bias is the significant computation time needed to explore the parameter space of a model. Here, we presented an ABC algorithm that allowed to partially solve this problem, although the number of runs were still in the high hundreds of thousands requiring the use of HPC resources.

A second obstacle is related to the nature of the archaeological data we used. Its high dimensionality, the noise and the low resolution of the dataset made summarising the data and comparing it with the simulation a challenge. In the trade-off between summarising the data and losing information we opted to keep as much information as possible with two very different patterns unfolding over long time periods while at the same time trying to fit a single general model to it. To do this, the function we used had to integrate at the same level very different time periods that may have been driven by very different processes, meaning that no one model would be able to match the data correctly. For example, we had to average periods

when four wares were competing against each other with periods where just two different wares were present.

This global approach means that the ABC algorithm would promote a model able to cope with the very general trends in the data, rather than particular smaller-scale aspects of the data pattern. The recommendation we made above for future work to focus on shorter time periods marked by the rapid rise and decline of a particular ware should be understood from this methodological perspective as well, as it is more likely to represent a coherent phenomenon rather than an aggregated average of multiple processes. The ABM + ABC method can easily be used at any scale, to understand what processes may be behind the distribution of goods at each individual time period, as we suggested for ITS. Without a doubt the key hindrance here is the access to adequately high resolution data. However, with the rise of open data in archaeology the hope is that currently non-interoperable or even not freely available archaeological datasets will be integrated to form a useful resource for this line of research.

Finally, this study demonstrates how computational modelling combined with ABC and a robust archaeological data pattern can make highly constructive contributions to theory evaluation, building and specification in Roman Studies. Cultural evolution studies are a source of well-defined formal models of interactions with a high potential to shed light on the impact of individual behaviours on complex, long-term processes such as ancient trade [57]. Here, we demonstrate how they can be used in conjunction with archaeological data. In this paper, we present an approach that explores theories addressing archaeological research questions, informed by a large archaeological data set. Such question- and data-driven selection of methodological and theoretical frameworks should become more widely applied in the historical disciplines and particularly in Roman Studies, where computational modelling is exceptionally rare [58]. We can similarly highlight the potential of archaeological data and research contexts to test social learning hypotheses at large temporal and geographical scales, which may enrich Cultural Evolution studies.

## Supporting information

**S1 Fig. Posterior distribution.** Complete posterior distributions for all parameters of the model. (TIF)

**S2 Fig. Simulation without ITS.** Comparison of the posterior for unbiased social learning with and without Italian Sigillata. The two experiments didn't present any differences. (TIF)

**S1 File. Source code & data.** The full project, with the data, the code of all components used to run the simulation and analyses is registered: https://osf.io/s5mdw/. (GZ)

## Acknowledgments

We thank Jeroen Poblome and Andrew Wilson for advice on tableware data and the Roman economy. We thank Philip Bes, Rinse Willet and Jeroen Poblome for the development of the open access ICRATES database.

## Author Contributions

**Conceptualization:** Simon Carrignon, Tom Brughmans.

**Data curation:** Tom Brughmans, Iza Romanowska.

**Formal analysis:** Simon Carrignon.

**Funding acquisition:** Iza Romanowska.

**Investigation:** Simon Carrignon.

**Methodology:** Simon Carrignon, Tom Brughmans, Iza Romanowska.

**Software:** Simon Carrignon.

**Supervision:** Tom Brughmans.

**Validation:** Simon Carrignon.

**Visualization:** Simon Carrignon.

**Writing – original draft:** Simon Carrignon, Tom Brughmans, Iza Romanowska.

**Writing – review & editing:** Simon Carrignon, Tom Brughmans, Iza Romanowska.

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
