## [Decision Letter · Decision Letter 0]

8 Jul 2020

PONE-D-20-15294

Tableware trade in the Roman East: exploring cultural and economic transmission with agent-based modelling and approximate Bayesian computation

PLOS ONE

Dear Dr. Carrignon,

Thank you for submitting your manuscript to PLOS ONE. After careful consideration, we feel that it has merit but does not fully meet PLOS ONE’s publication criteria as it currently stands. Therefore, we invite you to submit a revised version of the manuscript that addresses the points raised during the review process.

As you can see below, the two reviewers delivered complementary reports. While both of them raised some concerns about the assumptions on which your analysis is based, they focused their additional comments on different aspects. Reviewer 1 requested more information on the problem addressed in your work. On the contrary, Reviewer 2 asked for a clearer description of the methodology. These two issues are very important (even more considering the crossdisciplinary nature of the article) so, please, pay special attention to them when revising the manuscript.

Other comments made by the reviewers can also contribute to improve your work. In particular, those of reviewer 1 on data and code availability are especially relevant to the journal.

We look forward to receiving your revised manuscript.

Kind regards,

Sergi Lozano

Academic Editor

PLOS ONE

Journal Requirements:

2.  In your manuscript, please provide additional information regarding the specimens used in your study. Ensure that you have reported specimen numbers and complete repository information, including museum name and geographic location.

For more information on PLOS ONE's requirements for paleontology and archaeology research, see https://journals.plos.org/plosone/s/submission-guidelines#loc-paleontology-and-archaeology-research

Reviewers' comments:

Reviewer's Responses to Questions

**Comments to the Author**

1. Is the manuscript technically sound, and do the data support the conclusions?

Reviewer #1: Partly

Reviewer #2: Yes

2. Has the statistical analysis been performed appropriately and rigorously? 

Reviewer #1: Yes

Reviewer #2: Yes

3. Have the authors made all data underlying the findings in their manuscript fully available?

Reviewer #1: No

Reviewer #2: Yes

4. Is the manuscript presented in an intelligible fashion and written in standard English?

Reviewer #1: No

Reviewer #2: Yes

5. Review Comments to the Author

Reviewer #1: This is a very interesting paper, and the authors should be commended for using this approach with this data set, and we can hope that such work inspires the further use of computational methods in such contexts.

There are, however, some problems:

The whole paper needs another pass for English grammar, including use of apostrophes, spelling and pluralisation. The authors would do themselves a disservice by not improving this.

The data upon which the paper is based is not included in the publication, but is instead available from the original source: https://archaeologydataservice.ac.uk/archives/view/icrates_lt_2018/index.cfm This is different from including the data in the publication itself and should be explained.

The source code of the model itself is not clear: https://github.com/simoncarrignon/abcpandora leads to a “page not found” message and https://github.com/simoncarrignon/ceeculture is a large repository with no recent commits or clear structure. If the authors intend for others to use these tools in the way that they are used here, then they need to provide documentation.

Regarding the paper itself, first, some minor points, and then two general points that would help the paper:

Line 26: Give the dates instead of DATE A and DATE B, as this is the first time this appears in the text and the reader doesn't yet know what the authors are referring to.

In figure 1, the authors might want to show the area of origin for ITS. This would emphasise that it is coming from much further away.

Line 70 and line 292: The use of Pattern A and Pattern B is not clear. It appears that in the model, two different measures are used for pottery type – pattern A if the settlement includes one of the 5 main pottery types, or Pattern B if it does not. Does this mean that settlements with none of the of 5 main types are included in the model, and if so, how is their diversity defined? Why not either use only sites with one of the 5 main types or use a diversity measure for all sites?

Line 111: The authors state that taphonomic issues prevent the analysis of pottery frequency, and therefore eliminate any analysis that relies on falloff models or other density-dependent tests. Taphonomic issues are often used to dismiss all quantitative analysis of ancient artefacts. The authors are advised to explain more clearly why presence/absence is reliable here but frequency is not.

Line 194 and line 257 onwards: How does the model change with greater or lesser estimates of population (500)? What effect does this have on the outcome, or does it only increase the processing time?

Page 10: Please repeat the parameter definitions from table 2 here in table 3, so that the reader doesn't have to go back to work out what each of these values represent.

Line 348: Please define posterior distribution in this context before discussing its values.

Line 339: Isn't independent learning 23 times more likely than success-biased, according to table 5, not 20 times as in the text?

Line 422: This low rate of activity is an interesting result. Is there anything in the literature that this estimate can be compared with? Any Roman text source or description?

This last comment leads to my first general point:

It would be helpful to set up the problem more clearly within the context of the literature on Roman ceramics. How has this problem been defined within that literature? What solutions have been proposed? If the problem is not articulated, why is that, within this tradition? Including this information in the introduction would help to attract researchers interested in the topic but not the methodology, even if they don't agree with the outcome the authors propose. For example, pottery as a "space-filler" alongside more important cargo versus pottery as a valued export commodity in itself has been debated extensively in Attic figure painted pottery from a few centuries previous, with various evidence presented on both sides.

Second, and finally:

In line 79, the paper is talking about the strategies of traders, but in line 194, the agents are the settlements, and then in lines 369 onwards, the agents are described as individuals buying at the market. Based on the model, it appears the agents are the settlements, but this definition needs precision. In the conclusion on page 12, we're back to the strategies of traders, but this doesn't seem to be what the model is testing. The agents appear to be the settlements themselves, choosing which pottery type to consume at each time step, according a given strategy. A model of trader strategies within a population of settlements would be a very different system than that which is presented here.

Furthermore, if the pots are space-fillers alongside other cargo, then would the settlements have any agency at all? Is the behaviour described by this model a secondary effect from another market, such as that of raw materials discussed from line 396 onwards? In that case, what we see in these settlements is the material that gets sent on alongside these other shipments. This would then end up looking like Independent Learning on the part of the settlement, as the settlement is not really choosing, they're taking whatever turns up. This strategy requires no information, because the settlements can't control what gets put onto the ships that arrive. This needs to be explored further in the text.

There appears to be nothing wrong with the analysis itself, but the terms need to be defined more tightly, and the introduction, discussion and conclusion require further expansion.

Reviewer #2: This paper presents a case study of using Approximate Bayesian Computing to model trade of Roman tableware in the Mediterranean. It is building upon earlier work by the authors, but extends this by including a larger geographical region and validating the results against the available data, using ABC methods to allow modelling of a huge number of scenarios. The paper is clear about its goals and presents the outcomes well. Robustness of the outcomes seems guaranteed through the use of the methods applied, but reproducibility may be problematic for researchers who don't have access to high performance computing facilities.

There are two aspects of the paper that in my opinion need more attention before publication: 1) model assumptions and 2) technical background

Model assumptions:

The model implicitly assumes fully free trade, but what archaeological and historical evidence is underpinning this assumption? Without wanting to rekindle the primitivist-modernist debate, could other options be valid as well, like state-driven monopolies or trade (partly) regulated by conglomerates? You already indicate the possibility of tableware trade piggy-backing on bulk goods trade in the conclusions. Please provide more background on this debate, and why your angle is one that will bring a fresh view. This also applies to the social learning strategies, that are presented as if they are the only possible options. From the text, it is not even very clear what a 'buying strategy' is, or how you define a trader. Later, they are treated as urban centres that act independently. Again, some more underpinning of these theoretical points is needed.

Technical background:

I found sections 2-4 on the whole too condensed to easily follow the argument. The earlier models are referred to, but not properly summarized, so for new readers this will be very difficult to understand without reading the earlier papers. The pseudo-code is really too abstract for this. Then, the explanation of the ABC methods left me somewhat in the dark about what it actually does, even when the rationale is quite clearly stated. The nature of the epsilon-value, which is crucial to understanding why this method is better than running a full parameter sweep, should be much better explained. Please provide a more accessible introduction to the method - you probably don't need to convince people who already know how these things work. There is also some ambiguity in the formulation. In table 2.2 it is stated that initial values are inferred through ABC, but later it is said that they are hypothesized or sampled from a distribution. In section 4.1 and table 5 the numbers don't match (1.96 - twice and 23 - 20). Finally, give some more consideration to what it means to validate to the data in this context, especially since in section 2 you indicated that the 'data' are actually probabilities of tableware showing up in a site at a particular point in time, with only one-third of those sites actually being real sites with data. I would imagine that this will make things even more complicated.

6. PLOS authors have the option to publish the peer review history of their article (what does this mean?). If published, this will include your full peer review and any attached files.

Reviewer #1: No

Reviewer #2: No

---

## [Author Response · Author response to Decision Letter 0]

13 Aug 2020

Response to Reviewers

EDITOR:

As you can see below, the two reviewers delivered complementary reports. While both of them raised some concerns about the assumptions on which your analysis is based, they focused their additional comments on different aspects. Reviewer 1 requested more information on the problem addressed in your work. On the contrary, Reviewer 2 asked for a clearer description of the methodology. These two issues are very important (even more considering the crossdisciplinary nature of the article) so, please, pay special attention to them when revising the manuscript.

Other comments made by the reviewers can also contribute to improve your work. In particular, those of reviewer 1 on data and code availability are especially relevant to the journal.

Reply: Dear editor. We are very grateful for your work on this paper and for the highly constructive reviews provided by reviewers 1 and 2. In light of these we have made major changes to our paper, and we have addressed all comments by the reviewers. Please find below our reply to each of the points raised. We have also made all data and code/tools used for this work available in an open access repository (Open Science Framework). We have significantly elaborated on the substantive motivation and context for this work in Roman Studies. We have also elaborated the methodological sections, to make our description of our method and the concepts used much clearer. We believe this paper is now much stronger thanks to the feedback from the reviewers and yourself, and we are very grateful for this. We look forward to working more with you on this paper. Kind regards, Simon Carrignon, Tom Brughmans, Iza Romanowska

Reviewer #1:

This is a very interesting paper, and the authors should be commended for using this approach with this data set, and we can hope that such work inspires the further use of computational methods in such contexts.

There are, however, some problems:

The whole paper needs another pass for English grammar, including use of apostrophes, spelling and pluralisation. The authors would do themselves a disservice by not improving this.

Reply: Thank you, we have performed a thorough language check of the entire paper. Please note the many small language changes were not marked with track changes to ensure our major changes mentioned below are more clearly highlighted

The data upon which the paper is based is not included in the publication, but is instead available from the original source: https://archaeologydataservice.ac.uk/archives/view/icrates_lt_2018/index.cfm This is different from including the data in the publication itself and should be explained.

The source code of the model itself is not clear: https://github.com/simoncarrignon/abcpandora leads to a â�œpage not foundâ�� message and https://github.com/simoncarrignon/ceeculture is a large repository with no recent commits or clear structure. If the authors intend for others to use these tools in the way that they are used here, then they need to provide documentation.

Reply: in order to answer to the two previous comments and improve the reproducibility of the whole project, we decided to upload all the tools used in on the Open Science Framework website here: https://osf.io/s5mdw/files/?view_only=003bf28407694ad785c47c0733bd20cb Note that as the project contains multiple components you will need to download all of them separately and unzip all in a unique folder. The steps are described in the wiki of the osf project here: https://osf.io/s5mdw/wiki/home/?view_only=003bf28407694ad785c47c0733bd20cb . The project will be made public on acceptance of the manuscript.

Regarding the paper itself, first, some minor points, and then two general points that would help the paper:

Line 26: Give the dates instead of DATE A and DATE B, as this is the first time this appears in the text and the reader doesn't yet know what the authors are referring to.

Reply: thank you for pointing this out! This was placeholder text we forgot to update, we have now added the precise dates.

In figure 1, the authors might want to show the area of origin for ITS. This would emphasise that it is coming from much further away.

Reply: (we assume this comment refers to figure 2 rather than figure 1) thank you for this valuable suggestion. We have opted to modify figure 2, to state explicitly that ITS derives from Italy and Southern France.

Line 70 and line 292: The use of Pattern A and Pattern B is not clear. It appears that in the model, two different measures are used for pottery type â�“ pattern A if the settlement includes one of the 5 main pottery types, or Pattern B if it does not. Does this mean that settlements with none of the of 5 main types are included in the model, and if so, how is their diversity defined? Why not either use only sites with one of the 5 main types or use a diversity measure for all sites?

Thank you, we agree this needed to be explained much more clearly. Both patterns are used in the model as it says in the Equation (2). Indeed the settlements with none of the 5 types are used to test the model; the percentage they represent is shown by the lightest blue in the Fig 1. We modified the text to make that clearer in the legend of Fig 1 and added more explanation and an example when explaining the Equations 1 and 2. We could have used a measure of diversity for all settlements, and indeed started working with the Simpson diversity index, to try to reproduce the change observed in the distribution of this diversity index through time. But this would have allowed us to study only a general increase and decreasing of diversity, without taking into account which specific wares are responsible for those changes. This is why we coupled the metrics computed on PATTERN B (which measure the diversity of each settlement) with PATTERN A (which measures the geographical spread of each ware).

Line 111: The authors state that taphonomic issues prevent the analysis of pottery frequency, and therefore eliminate any analysis that relies on falloff models or other density-dependent tests. Taphonomic issues are often used to dismiss all quantitative analysis of ancient artefacts. The authors are advised to explain more clearly why presence/absence is reliable here but frequency is not.

Reply: We thank the reviewer for this comment, and agree with the need to be more explicit about this assumption. We have now added an elaboration to section 2.1 that argues the practice of only recording and publishing diagnostic sherds, and the ICRATES database being an aggregation of this practice, makes the volumes in this database not representative of excavated pottery counts. We also argue that diversity of non-local wares are a reliable pattern in the database we use.

Line 194 and line 257 onwards: How does the model change with greater or lesser estimates of population (500)? What effect does this have on the outcome, or does it only increase the processing time?

Reply: This is indeed an important remark, thank you. Previous work (Gintis 2006,2007, Carrignon 2015) has not noticed much effect of the population size once the number of agents is much bigger than the number of goods exchanged. It may accelerate the reaching of an equilibrium in some cases, but the system in this paper changes relatively quickly, which should even minimise the impact of the population size. We performed some preliminary tests that didn't show much difference between 250 and 2000 agents, but haven't included them here as they were done on a theoretical and preliminary version of the model. Moreover, as the reviewer suggests, the main impact of increasing the population size is increasing non linearly the computation time, which totally prevented us from trying the Approximate Bayesian Computation with different numbers of agents and choose what seemed to us an appropriate trade-off between computation time and historical knowledge.

Page 10: Please repeat the parameter definitions from table 2 here in table 3, so that the reader doesn't have to go back to work out what each of these values represent.

Reply: We have now repeated the descriptions from table 2 in table 3.

Line 348: Please define posterior distribution in this context before discussing its values.

Reply: We added clarifications about what the posterior distributions are in this paragraph

Line 339: Isn't independent learning 23 times more likely than success-biased, according to table 5, not 20 times as in the text?

Reply: yes, the Bayes factor is equal to ~ 23.5, we corrected it in the text. We initially rounded the results because the Bayes factors are meaningful when they differs for multiple orders of magnitude, but we agree with this reviewer this should be 23.5

Line 422: This low rate of activity is an interesting result. Is there anything in the literature that this estimate can be compared with? Any Roman text source or description?

Reply: Thank you for raising this question. We have added sources and text to emphasis the role of travel time and travel conditions, which make infrequent long-distance interactions of a few times a year rather likely. We specified that in our model this result refers to long-distance economic interactions only.

This last comment leads to my first general point:

It would be helpful to set up the problem more clearly within the context of the literature on Roman ceramics. How has this problem been defined within that literature? What solutions have been proposed? If the problem is not articulated, why is that, within this tradition? Including this information in the introduction would help to attract researchers interested in the topic but not the methodology, even if they don't agree with the outcome the authors propose. For example, pottery as a "space-filler" alongside more important cargo versus pottery as a valued export commodity in itself has been debated extensively in Attic figure painted pottery from a few centuries previous, with various evidence presented on both sides.

Reply: We thank the reviewer for pointing out the need for this. In our earlier manuscript, we clearly tried to keep the Roman ceramics section very concise in comparison to the description of our methodological work. We have now significantly expanded section 1.1 with a discussion of how the ceramic data patterns have been discussed in the Roman ceramics literature, what explanatory factors were proposed, and why we decided to explore our three hypotheses.

Second, and finally:

In line 79, the paper is talking about the strategies of traders, but in line 194, the agents are the settlements, and then in lines 369 onwards, the agents are described as individuals buying at the market. Based on the model, it appears the agents are the settlements, but this definition needs precision. In the conclusion on page 12, we're back to the strategies of traders, but this doesn't seem to be what the model is testing. The agents appear to be the settlements themselves, choosing which pottery type to consume at each time step, according a given strategy. A model of trader strategies within a population of settlements would be a very different system than that which is presented here.

Reply: We thank the reviewer for his comment as the terminology was indeed confusing. We tried to change it throughout the text and stated more explicitly what we intended by agents.

Furthermore, if the pots are space-fillers alongside other cargo, then would the settlements have any agency at all? Is the behaviour described by this model a secondary effect from another market, such as that of raw materials discussed from line 396 onwards? In that case, what we see in these settlements is the material that gets sent on alongside these other shipments. This would then end up looking like Independent Learning on the part of the settlement, as the settlement is not really choosing, they're taking whatever turns up. This strategy requires no information, because the settlements can't control what gets put onto the ships that arrive. This needs to be explored further in the text.

Reply: The reviewer's description that the space-filler hypothesis might indeed look like independent learning in this model. This is why we posit this in section 5.1 as something to be explored in more detail in future work. However, we have to emphasise that we decided not to make these assumptions the starting point of our study because: (1) the piggy-back trade based on other goods is a hypothesis the effects of which on distributions of tableware need to be studied in future simulation studies, which is what we hope our results will encourage, and (2) some authors argue explicitly that the wide distribution of tableware and the differences in wares' distributions suggests access to commercial innovation which is not captured by the independent learning hypothesis (hence we tested three hypotheses). The reviewer's comment makes it clear to us that we need to be more transparent and explicit in why we discuss the space-filler/piggy-back hypothesis in the context of our model and results. We have made additions to section 5.1 in light of this, and we thank the reviewer for expressing their interpretation of this phenomenon so clearly, which has very much helped us in formulating these changes to the manuscript. 

There appears to be nothing wrong with the analysis itself, but the terms need to be defined more tightly, and the introduction, discussion and conclusion require further expansion.

Reply: we thank the reviewer for their thorough engagement with our work, and for the constructive detailed feedback. We have incorporated all recommendation and made corrections for mistakes pointed out by the reviewer. We believe the manuscript is much stronger because of it, for which we are grateful to the reviewer.

** Reviewer #2: 

This paper presents a case study of using Approximate Bayesian Computing to model trade of Roman tableware in the Mediterranean. It is building upon earlier work by the authors, but extends this by including a larger geographical region and validating the results against the available data, using ABC methods to allow modelling of a huge number of scenarios. The paper is clear about its goals and presents the outcomes well. Robustness of the outcomes seems guaranteed through the use of the methods applied, but reproducibility may be problematic for researchers who don't have access to high performance computing facilities.

There are two aspects of the paper that in my opinion need more attention before publication: 1) model assumptions and 2) technical background

Model assumptions:

The model implicitly assumes fully free trade, but what archaeological and historical evidence is underpinning this assumption? Without wanting to rekindle the primitivist-modernist debate, could other options be valid as well, like state-driven monopolies or trade (partly) regulated by conglomerates? You already indicate the possibility of tableware trade piggy-backing on bulk goods trade in the conclusions. Please provide more background on this debate, and why your angle is one that will bring a fresh view. This also applies to the social learning strategies, that are presented as if they are the only possible options. From the text, it is not even very clear what a 'buying strategy' is, or how you define a trader. Later, they are treated as urban centres that act independently. Again, some more underpinning of these theoretical points is needed.

Reply: We entirely agree with the reviewer that we did not underpin our theoretical assumptions explicitly and elaborately enough. We have made significant changes and additions to sections 1.1 and 5.1 of the text and added references to relevant literature.

The reviewer is right to point out this assumption, although the hypotheses we test radically change the nature of this free trade: free trade informed by complete knowledge (success-bias), informed by some knowledge (unbiased), informed by no knowledge (independent learning). Nevertheless, each agent is able to obtain all non-local products in theory although in practice realistic simulation experiments such as those underlying our results show distributions closer to the data.

What concerns the reviewer's comment about terminology: we agree and we have made changes throughout to make this terminology more clear and less ambiguous. Moreover, we have now stated explicitly at the first mentioning of our agents in section 2.2 what an agent represents in our model.

Technical background:

I found sections 2-4 on the whole too condensed to easily follow the argument. The earlier models are referred to, but not properly summarized, so for new readers this will be very difficult to understand without reading the earlier papers. The pseudo-code is really too abstract for this.

Reply: We agree with the reviewer that understanding the model was very hard from our previous manuscript. This was the case because we limited our description to the essentials, as the model was already described in length in previous papers. Nonetheless, we have now significantly expanded our description of the model and other methodological aspects in sections 2-4, which we agree makes the method easier to follow.

Then, the explanation of the ABC methods left me somewhat in the dark about what it actually does, even when the rationale is quite clearly stated. The nature of the epsilon-value, which is crucial to understanding why this method is better than running a full parameter sweep, should be much better explained. Please provide a more accessible introduction to the method - you probably don't need to convince people who already know how these things work. 

Reply: As per our previous reply, we understand that the technique wasn't easy to follow as we limited it's description to the bare minimum. To remedy this, we added several paragraphs to clarify the sequential nature of the ABCPMC algorithm in section 3.1.

There is also some ambiguity in the formulation. In table 2.2 it is stated that initial values are inferred through ABC, but later it is said that they are hypothesized or sampled from a distribution. 

Reply: The notation was indeed confusing. By "inferred" in table 2 we meant that these were the parameters explored through ABC, which implies they will be sampled from a distribution. We changed that in table 2 and added explanation in the legend.

In section 4.1 and table 5 the numbers don't match (1.96 - twice and 23 - 20). Finally, give some more consideration to what it means to validate to the data in this context, especially since in section 2 you indicated that the 'data' are actually probabilities of tableware showing up in a site at a particular point in time, with only one-third of those sites actually being real sites with data. I would imagine that this will make things even more complicated.

Reply: We thank the reviewer for this comment, because it reveals we were not clear enough on what we tried to achieve with Bayesian inference. As the reviewer writes, things are more much more complicated and given the high level of uncertainty it's hard to "validate" our model and remove any reference to it in the text. The Bayes Factor helps us see which model is more likely to reproduce the data. This is a probabilistic approach that doesn't give absolute quantities that validate a model, but a qualitative relation between the likelihood of different models to reproduce the data. This is why we first approximated the data in the text as the reviewer noted. In section 4.1 we made changes to avoid confusion, and added a sentence to emphasise this nature of the factors calculated. We also changed the caption of table 5.

---

## [Decision Letter · Decision Letter 1]

3 Sep 2020

PONE-D-20-15294R1

Tableware trade in the Roman East: exploring cultural and economic transmission with agent-based modelling and approximate Bayesian computation

PLOS ONE

Dear Dr. Carrignon,

Thank you for submitting your manuscript to PLOS ONE. As you can see below, both reviewers are satisfied with the current version. However, they also noticed some typos in the text.

Since PLOS ONE does not copyedit manuscripts to be published, I suggest you check your manuscript for language usage, spelling, and grammar. If you do not know anyone who can help you do this, you may wish to consider employing a professional scientific editing service.

We look forward to receiving your revised manuscript.

Kind regards,

Sergi Lozano

Academic Editor

PLOS ONE

Reviewers' comments:

Reviewer's Responses to Questions

**Comments to the Author**

1. If the authors have adequately addressed your comments raised in a previous round of review and you feel that this manuscript is now acceptable for publication, you may indicate that here to bypass the “Comments to the Author” section, enter your conflict of interest statement in the “Confidential to Editor” section, and submit your "Accept" recommendation.

Reviewer #1: All comments have been addressed

Reviewer #2: All comments have been addressed

2. Is the manuscript technically sound, and do the data support the conclusions?

Reviewer #1: Yes

Reviewer #2: Yes

3. Has the statistical analysis been performed appropriately and rigorously? 

Reviewer #1: Yes

Reviewer #2: Yes

4. Have the authors made all data underlying the findings in their manuscript fully available?

Reviewer #1: Yes

Reviewer #2: Yes

5. Is the manuscript presented in an intelligible fashion and written in standard English?

Reviewer #1: Yes

Reviewer #2: Yes

6. Review Comments to the Author

Reviewer #1: The paper is now much clearer, especially with regard to the units of analysis. Due to time limitations, I was not able to check the code, but I look forward to doing so when this is published.

If possible, please give the changed sections another pass for English sense and case. There is a major typo on line 582 in the text with track changes enabled: "exited" should be "existed". Please change this.

Thank you for the opportunity to review this paper.

Reviewer #2: The revised paper has adequately addressed all my comments. I still noticed a few typos ('notted' in line 281; 'a prior distribution that describe' in line 320), please give it a thorough last look before final publication.

7. PLOS authors have the option to publish the peer review history of their article (what does this mean?). If published, this will include your full peer review and any attached files.

Reviewer #1: No

Reviewer #2: No

---

## [Author Response · Author response to Decision Letter 1]

25 Sep 2020

Dear editor,

We updated the manuscript and the latex source file, correcting the typos mentioned by the reviewers and other that we noted after careful examination. We may have not fully updated the file with the track changes enables, as changes were minor typos and we understood that this version won't be send to the reviewers anymore, some changes may be missing in this version. If you want us to integrate all our last minor edits to the version with track changes, let us know and we will do so as soon as possible.

---

## [Editor Report · Decision Letter 2]

28 Sep 2020

Tableware trade in the Roman East: exploring cultural and economic transmission with agent-based modelling and approximate Bayesian computation

PONE-D-20-15294R2

Dear Dr. Carrignon,

We’re pleased to inform you that your manuscript has been judged scientifically suitable for publication and will be formally accepted for publication once it meets all outstanding technical requirements.

Kind regards,

Sergi Lozano

Academic Editor

PLOS ONE
---

## [Editor Report · Acceptance letter]

13 Nov 2020

PONE-D-20-15294R2 

Tableware trade in the Roman East: exploring cultural and economic transmission with agent-based modelling and approximate Bayesian computation 

Dear Dr. Carrignon:

I'm pleased to inform you that your manuscript has been deemed suitable for publication in PLOS ONE. Congratulations! Your manuscript is now with our production department. 

Kind regards, 

on behalf of

Dr. Sergi Lozano 

Academic Editor

PLOS ONE